# Men and sexual and reproductive healthcare in the Nordic countries: a scoping review

Mazen Baroudi ,[1] Jon Petter Stoor,[1,2] Hanna Blåhed,[1] Kerstin Edin,[1] Anna-Karin Hurtig[1]

¹Department of Epidemiology and Global Health, Umeå University Faculty of Medicine, Umea, Sweden
²Centre for Sami Health Research, Department of Community Medicine, UiT The Arctic University of Norway, Tromso, Norway

**Correspondence to**
Dr Mazen Baroudi;
mazen.baroudi@umu.se

## ABSTRACT

**Context** Men generally seek healthcare less often than women and, other than traditional gender norms, less is known about the explanation. The aim was to identify knowledge gaps and factors influencing men regarding sexual and reproductive healthcare (SRHC) in the Nordic countries.

**Methods** We searched PubMed and SveMed+ for peer-reviewed articles published between January 2010 and May 2020. The analyses identified factors influencing men's experiences of and access to SRHC.

**Results** The majority of the 68 articles included focused on pregnancy, birth, infertility and sexually transmitted infections including HIV. During pregnancy and childbirth, men were treated as accompanying partners rather than individuals with their own needs. The knowledge and attitudes of healthcare providers were crucial for their ability to provide SRHC and for the experiences of men. Organisational obstacles, such as women-centred SRHC and no assigned healthcare profession for men's sexual and reproductive health issues, hindered men's access to SRHC. Lastly, the literature rarely discussed the impact of health policies on men's access to SRHC.

**Conclusions** The literature lacked the perspectives of specific groups of men such as migrants, men who have sex with men and transmen, as well as the experiences of men in SRHC related to sexual function, contraceptive use and gender-based violence. These knowledge gaps, taken together with the lack of a clear entry point for men into SRHC, indicate the necessity of an improved health and medical education of healthcare providers, as well as of health system interventions.

## STRENGTHS AND LIMITATIONS OF THIS STUDY

⇒ This review is the first to examine the experiences of men in sexual and reproductive healthcare in the Nordic countries.
⇒ We used of a Nordic-specific database without restriction to language.
⇒ Search was restricted to two databases but complemented with manually screening the reference lists of the identified literature.
⇒ The broad nature of the field and the wide variety of terms related to sexual and reproductive health make it difficult to assure the inclusion of all relevant literature.
⇒ We implicitly treated the Nordic countries as essentially similar, which might obscure important differences between or within countries.

## INTRODUCTION

Addressing men's sexual and reproductive health (SRH) needs alongside that of women's is essential, however men's SRH is neglected. Men generally seek healthcare, especially primary healthcare, to a lower degree than women, and this also applies to sexual and reproductive healthcare (SRHC).[1 2] For example, 23% of over 40-year-old men in Europe reported sexual dysfunction but only one-quarter of them sought healthcare,[3] and similar results have been reported in Sweden.[4] Also, studies from Sweden and Norway have indicated that youth clinics are perceived as 'women clinics'. Therefore, fewer men seek these services compared with young women.[5 6] Additionally, although not universal, men test themselves for sexually transmitted diseases to a lower extent compared with women.[7 8] However, various groups of men might have different health-seeking behaviours and different experiences in SRHC. For example, men with high socioeconomic status[1 9] and men who have sex with men (MSM) seek SRHC more often.[10 11] The higher use of SRHC among MSM might be due to their higher needs. Furthermore, MSM experiences in SRHC might differ due to their level of openness about their sexual orientation and due to structural factors such as homophobia.[10 11]

Traditional gender norms might urge men to be independent, strong and invulnerable and also hinder them from acknowledging having problems, creating a barrier to seeking healthcare.[12] In particular, admitting sexual health problems might imply more vulnerability for men, thus decreasing the likelihood of seeking healthcare.[5 13] Even though gender norms play an important

role in men's health-seeking behaviours, it cannot alone explain the lower utilisation of SRHC. Men who eventually sought SRHC did not get the help they expected. For example, more than half of men who sought help related to sexual function in Sweden reported not getting enough support.[4] Furthermore, men often felt excluded in healthcare related to infertility and pregnancy.[14 15] These experiences might be due to the lack of response of the health system to men's needs that can be related to healthcare organisation and delivery,[9] including no support or guidelines for health professionals to promote men's SRH.[16] Additionally, health and medical education in Sweden, as an example, does not have enough focus on men's SRH.[16 17]

The Nordic countries are among the best in the world in the available international gender equality statistics.[18] Since gender inequality affects women's SRH to a larger degree compared with men,[19 20] there is a greater focus on women's rights to SRH. Men's SRH does not get the same attention in practice and little is known about men's SRH in the Nordic countries.[21] The available literature mainly focuses on gender norms and masculinities and its link to health-seeking behaviours and risk taking, while much less is known on how men are experiencing SRHC.[9 21–23]

## Aim

The aim of this scoping review was to identify knowledge gaps and factors influencing men regarding SRHC in the Nordic countries during the period between 2010 and 2020.

## METHOD

This review was performed according to Arksey and O'Malley's method stages for conducting a scoping review, which include identifying the research question, literature search, study selection, charting and synthesising.[24] The research questions included: (1) What is the current status of the literature published in Scandinavian regarding men and SRHC?; (2) How men in the Scandinavian countries are experiencing SRHC?

### Search strategies and selection criteria

A structured search of the literature was conducted using two databases, PubMed and SveMed+ (a Scandinavian database) without restriction of language. Search terms included sexual and reproductive health, men, healthcare, experiences and Nordic countries (see online supplemental appendix 1 for detailed search terms). The following eligibility criteria were used: (1) peer-reviewed empirical studies, all study designs were considered; (2) published between January 2010 and May 2020; (3) assessing men's experiences in SRHC or perspectives of healthcare providers (HCPs) on men's SRHC; and (4) conducted in the Nordic countries.

The initial search gave 1286 articles (896 from PubMed and 390 from SveMed+). After screening the titles and abstracts, 108 articles were read in full, and after being

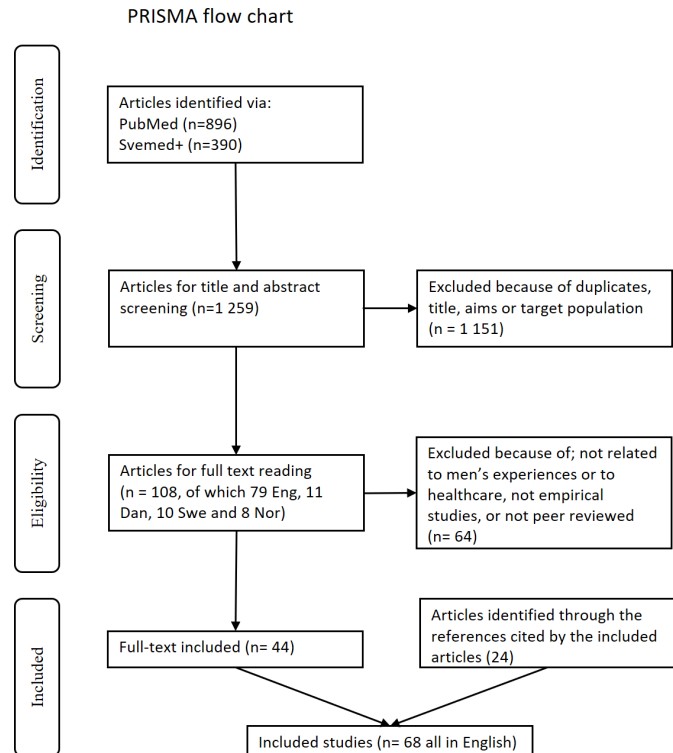

PRISMA flow chart

**Identification**

Articles identified via:
PubMed (n=896)
Svemed+ (n=390)

**Screening**

Articles for title and abstract screening (n=1 259)

Excluded because of duplicates, title, aims or target population (n = 1 151)

**Eligibility**

Articles for full text reading (n = 108, of which 79 Eng, 11 Dan, 10 Swe and 8 Nor)

Excluded because of; not related to men's experiences or to healthcare, not empirical studies, or not peer reviewed (n= 64)

**Included**

Full-text included (n= 44)

Articles identified through the references cited by the included articles (24)

Included studies (n= 68 all in English)

**Figure 1** Preferred Reporting Items for Systematic Reviews and Meta-Analyses (PRISMA) flow chart on search results of men's experiences in sexual and reproductive healthcare in the Nordic countries.

judged for their eligibility, 44 articles remained. Additional 24 papers were identified through the reference lists of these papers, resulting in 68 papers included in this scoping review (figure 1). The articles were judged for eligibility by the first author, but when uncertainties arose, two coauthors read and judged the articles for eligibility separately. The three researchers then discussed the articles and decided unanimously on the inclusion/exclusion of these articles.

### Data extraction and synthesis

The identified articles were mapped using the WHO framework for operationalising SRH,[25] and the result part of each article was extracted and coded using sensitising concepts of healthcare experiences (online supplemental appendix 2). Thereafter, the results were synthesised using a theoretical framework, adapted from Kilbourne *et al*, which provides health service research perspectives on understanding health and healthcare disparities.[26]

### Patient and public involvement

Patients and the public were not involved in this study.

## RESULTS

### Description of the identified studies

Despite not restricting the language of the studies, all the 68 studies included were in English. The absolute majority of the studies were conducted in Sweden (54 articles), while six studies were conducted in Denmark,

five in Norway and three in more than one country. No studies were identified from Iceland or Greenland.

Half of the studies (34) adopted a qualitative design, 32 studies a quantitative design and 2 studies a mixed-methods design. Most of the studies (61 articles) were about men's perspectives of SRHC, while seven studies covered the perspectives of HCPs. Of the studies dealing with men's perspectives, 16 studies assessed women's perspectives together with that of men. Apart from two articles about the experiences of transgender men, the articles did not mention gender identities. Most of the papers dealing with men's perspectives referred to the overall experience of healthcare and healthcare staff in general. Of the 28 papers referring to specific primary HCPs, 14 mentioned midwives, 8 mentioned physicians and 6 mentioned nurses.

SRH topics were grouped with help of the WHO framework for operationalising sexual health and its linkages to reproductive health.[25] This framework was used because it demonstrates the interlinked nature between sexual health and reproductive health, yet clearly distinguishes

topics for intervention and research in both sexual health and reproductive health (figure 2). Besides the eight topics from this framework, SRH cancers were also added, while the remaining studies with no one topic of focus were grouped under 'other'.

More than one-third of the papers were about the experiences of fathers/expectant fathers during antenatal, intrapartum and postnatal care (25 papers, including 12 about antenatal care and 11 about intrapartum care), while 15 papers dealt with sexually transmitted infections (STIs), mainly HIV (12 papers) and MSM (9 papers). We found 11 papers concerning men's experiences in infertility care (three of them were related to infertility among patients with cancer) and 11 papers in cancer care. We also found four studies dealing with sexual education and information (two of them related to cancer and the other two related to antenatal care), three studies about abortion care, two studies about sexual violence and two studies about sexual functioning and counselling (both related to patients with cancer). We found no study dealing

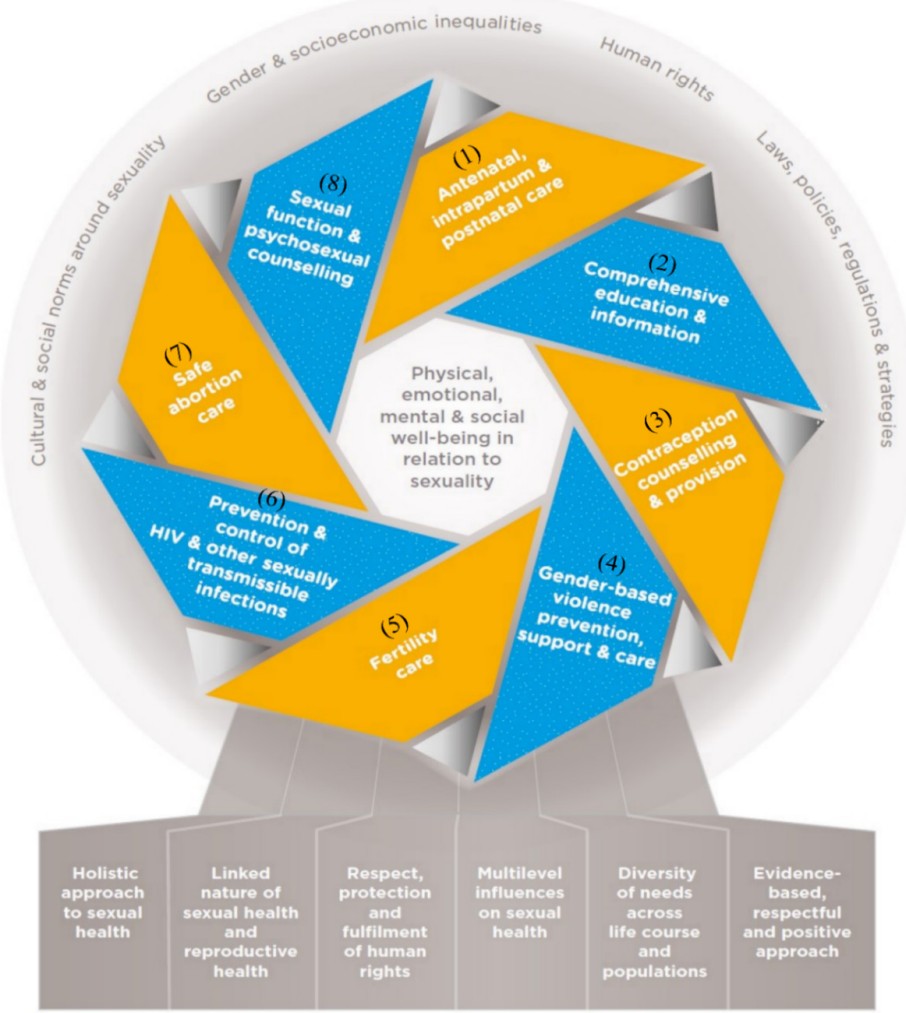

**Figure 2** Framework for operationalising sexual health and its linkages to reproductive health (from 'Sexual health and its linkages to reproductive health: an operational approach').[25] The intertwined blue and orange ribbons represent sexual health and reproductive health, respectively.

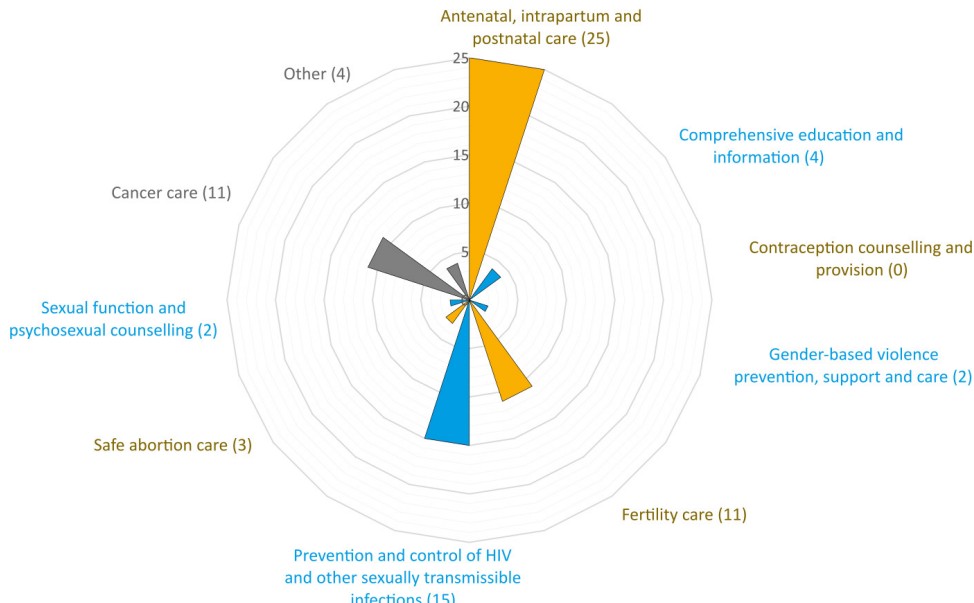

### Studies identified by sexual and reproductive health topic

**Figure 3** Men's experiences in sexual and reproductive healthcare in the Nordic countries. Number of studies identified grouped by sexual and reproductive health topics.

with the provision of men's contraceptive counselling (figure 3).

### Theoretical framework for analysis

The identified literature dealt with men's experiences in SRHC from various perspectives and can be organised in the framework adapted from Kilbourne et al.[26] Kilbourne et al framework provides a multilevel approach to understand healthcare disparities. It provides an ecological lens that goes beyond individual to interpersonal and organisational factors. The factors influencing men's experiences are divided into (1) individual, including HCPs and users; (2) interpersonal, which deals with the healthcare encounter and contact circumstances; (3) organisational, which deals with healthcare system

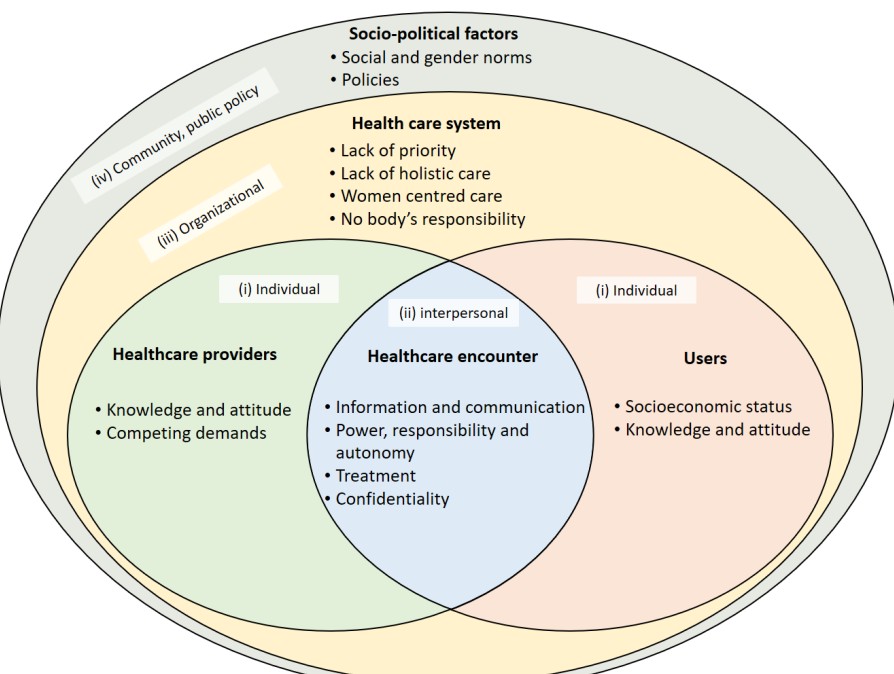

**Figure 4** Theoretical framework for analysis of men's experiences in sexual and reproductive healthcare, adapted from Kilbourne et al.[26]

factors; and (4) the larger influence of the community and public policies (figure 4).

## HCPs' factors

The literature described how factors related to HCPs, such as sex, attitude, knowledge and competence, affect the HCP–user relationship and experiences of men in healthcare. For example, female HCPs did not prevent men from talking about their concerns regarding infertility.[27] Similarly, men diagnosed with prostate cancer wished to talk about sexuality with a mature knowledgeable HCP, without considering their sex.[28] Furthermore, disclosing victimisation to female HCPs as compared with male HCPs was claimed to be easier for some men.[29]

### Varied levels of knowledge, competing demands and differing attitudes

Lacking knowledge about men's SRH was expressed by various HCP professions and were associated with less ability to deal with men's SRH consultations. For example, nurses perceived that their lack of knowledge was influencing their preparedness to provide sexual health consultations for men.[16] Midwives also expressed their limited knowledge about male SRH, which was considered essential if inviting men for SRH consultations.[17] Also, less experienced physicians (young and/or under training) felt uncomfortable dealing with sexual health consultations.[30] Moreover, the competing demands in the form of high workload and limited time hindered HCPs from discussing SRH with men.[16 31 32]

Differing attitudes towards health-seeking behaviours of men were found. While most HCPs were described as having positive attitudes, being friendly, sensitive and supportive,[33–36] some were still perceived as harsh and non-responsive.[33 34] These negative attitudes were sometimes perceived by men as discrimination based on their sex, which hindered them, for example, from disclosing victimisation.[29]

### The view of men in reproductive healthcare services, an accompanying partner or an individual?

Even though HCPs in reproductive healthcare services usually deal with couples having a common reason for visiting healthcare, in most cases, they have primarily communicated with the women.[27 33 37 38] Women were in focus during infertility treatment, pregnancy and birth, leading men to feel neglected, invisible and superfluous during the visits.[14 15 39 40] The lack of interest in listening to or interacting with men also hindered their involvement in supporting their partners, for example, when giving birth.[41 42]

The lack of focus on men might be explained by time constraints and no time being allocated to men's concerns during visits.[17] Anyhow, the attitudes and behaviours of HCPs generally made a difference in men's perception of their involvement or lack of involvement in healthcare.[39] Couples highlighted the need to treat partners on equal terms and to focus on them as a unit rather than solely on the women,[14 15 27 37] and they expected communication as inclusive with both partners.[36] Men also expressed that HCPs should welcome them to more active involvement during birth and support their role as expectant fathers. HCPs should acknowledge men's needs and give them the opportunity to talk about their concerns.[14 27 33 43]

Examples of good practices involving men in reproductive healthcare are also mentioned in the literature. One example was participatory parental classes or separate parental classes for men and women dealing with men's concerns related to pregnancy and birth, which helped men to take part and to feel involved.[40 44] Another example was assigning tasks and continuously informing men during labour and allowing the father to stay at the hospital after the baby is born. These practices gave men a feeling of being important and recognised, hence receiving needed support.[42 45–47]

## Healthcare users' factors

The literature described users' factors that influence their experiences in healthcare. This included men's socioeconomic situation, including education, age, knowledge and attitude. For example, the age of users were discussed in relation to the age of HCPs; nurses were more comfortable talking about sexuality with younger men as compared with men of their own age or older.[16] Young men, in comparison with young women, were pointed out as being less acquainted with youth clinics or where else to seek SRHC.[5]

Healthcare users' factors are discussed in the literature mainly in three SRH subject areas, namely, prevention and control of HIV and other STIs, antenatal/intrapartum care and cancer care, which is elaborated on below.

### Prevention and control of HIV and other STIs

Most of the literature focused on HIV testing, treatment and their sociodemographic determinants (see box 1 for more details about the factors discussed in the literature).

The literature relating to other STIs (besides HIV) was limited to the attitudes of upper secondary school boys (median age=18 years) toward human papillomavirus (HPV) vaccination and attitudes of men toward STI testing during the pregnancies of their partners. Upper secondary school boys had a positive attitude with regard to participation in HPV vaccinations; they stated that vaccinating only girls is unfair. Even though they had a positive attitude to share the responsibility of STI prevention, boys rarely used condoms, especially if they knew their sex partner in advance.[48] Men's attitudes toward STI testing during pregnancy were diverse. Some men perceived the test as an 'infidelity check' that is sensitive and can risk the relationship, while others perceived it as a safety measure that should be 'routine' during pregnancy.[13 49]

### Antenatal and intrapartum care

The literature discussed men's socioeconomic characteristics, knowledge and experiences in antenatal and

---

about the effects of cancer treatment on sexual desire, sexual function and fertility.[55]

**Box 1   Key characteristics of users in relation to HIV testing and treatment**

⇒ Age: younger age was reported to be associated with higher HIV testing among men who have sex with men (MSM)[75 76] and earlier diagnosis in the general population.[10]

⇒ Education: a lower level of education was associated with less testing for HIV in the general population,[75] but not among MSM.[76]

⇒ Country of birth: studies showed that country of birth was not associated with lower HIV testing among MSM.[76 78] However, two-thirds of the foreign-born patients with HIV had not been tested for HIV at migration to Sweden.[10 59] Therefore, foreign-born men were more likely to be diagnosed late (65% of foreign-born compared with 43% of Swedish-born) and less likely to optimally adhere to HIV treatment.[10 87]

⇒ Sexuality: since HIV testing was perceived as implicitly implying same-sex sexual relations, non-disclosing MSM were more likely to have never been tested for HIV.[75 77 88] However, MSM were less likely to be diagnosed late (40% of MSM compared with 67% of heterosexual patients) and less likely to optimally adhere to HIV treatment.[10 87]

⇒ Knowledge: men's knowledge about HIV transmission was associated with never being tested for HIV among MSM and the general population.[75 76] Never being tested for HIV was also associated with not knowing if the tests were free or affordable[75 89] and lack of knowledge about HIV testing services.[45 76 78] For example, only one-fourth of MSM knew about home sampling (internet-ordered tests),[90] and around 40% have never heard of the Testpoints programme (peer-led testing performed in MSM clubs, among other places).[91]

⇒ Risk perception: the perception of having a very low risk of contracting sexually transmitted infections (STIs), including HIV, was highly associated with never being tested for HIV or STIs.[3 7 49 78]

---

intrapartum healthcare. Lack of knowledge about antenatal services, such as antenatal classes, was common among men; they usually had not heard about the service before but received information from their partners.[37] Men who had no social support from family and friends during the pregnancies of their partners were more dissatisfied with antenatal care and less likely to attend parental classes.[50] Studies found younger age and higher education level were associated with lower satisfaction with the overall birth experience,[41 51] while no such association was reported in relation to men's country of birth.[52] Additionally, younger men as compared with older men, perceived midwives as less supportive, less attentive and as not inspiring confidence.[51] These differences might be explained by younger men having higher expectations.[53]

### Cancer care and SRH

The literature explored the factors of users related to cancer care, especially the effects of cancer treatment on fertility and sexuality. The majority of physicians claimed that they discussed the impact of cancer treatment on fertility if the patient was at reproductive age. However, one-third of the physicians did not do this regularly.[31 54] Around half of men in the 41–60 years old age group claimed that they had not received enough information

Similar to other SRH services, lack of knowledge about the services was common, with only around one-fifth of men knowing about the prostate-specific antigen test for prostate cancer screening before testing.[56] Studies showed no associations between age and the overall satisfaction with cancer care, while a higher level of education was associated with lower overall satisfaction with prostate cancer care.[57] Furthermore, the literature indicated that manual workers were less likely to receive a bone scan and radical prostatectomy, and they had higher overall and cancer-specific mortalities as compared with non-manual employees.[58]

### Healthcare encounter factors

The factors under which the healthcare encounters took place influenced the HCP–user relationship and experiences of users. The literature discussed, among other issues, HCP–user communication and the power and autonomy of men.

### Information and communication

Information and communication were recurring themes in all SRH subject areas. More than half of the studies touched on some aspect of information, or the way it is delivered and communicated. Receiving information was described as valuable and important and made men feel pleased, satisfied and empowered.[59–65] During the birth process, for example, information helped men to feel included and to find their place in supporting their partners and facilitated the decision-making of couples.[33 43 46 60] Contrarily, lack of sufficient information was associated with more concerns and feelings of exclusion and dissatisfaction.[41 66] Insufficient information was reported in various healthcare settings, for example, the effects of antenatal care,[37 67–69] infertility care[62] and cancer treatment/surgery on sexual health.[55 70]

The literature also discussed the format of information. Oral information was especially preferred when the matter aroused many questions, such as communicating an infertility diagnosis[36] or HPV vaccination,[48] while written information was considered more suitable in other cases, such as HIV and STI information for MSM.[63] However, even though recommended by the National Board of Health and Welfare in Sweden, studies have shown that the majority of men did not receive written information about prostate cancer screening and some were not even aware that they underwent the screening.[56] In other cases, a combination of oral and written information was considered easier to comprehend, for example, when communicating the side effects of cancer treatment on fertility[35] (see box 2).

### Lack of control and compromised autonomy in reproductive healthcare

Men's engagement in reproductive healthcare seemed to be a complex matter; midwives valued men's

**Box 2    The characteristics of satisfying information and communication—men's views**

⇒ Clear and simple language: clear and proper level information was perceived as important. The inability to understand the medical language of healthcare providers (HCPs) caused distress.[33 35 45 60]

⇒ Reliable: contradictions, unrealistic information and lack of reliable information caused frustration.[34 43] Exaggerated information (ie, understating or overstating the real situation) was associated with unease, confusion and a sense of not being taken seriously.[59] Men wanted to feel welcome when asking questions and wanted honest, consistent and clear answers.[42 43] Men expressed a need for help to choose reliable websites and organise and discuss the information received.[14]

⇒ Personalised and relevant: while general information could be obtained from the internet, receiving personalised and relevant information from the HCPs was a high priority.[27 47 60] For example, an online patient–nurse communication service played a central role in providing personalised information for patients with cancer.[92]

⇒ Comprehensive and sufficient: receiving adequate and comprehensive information was regarded as important.[59] For example, men highlighted the need for a deeper dialogue about personal experiences or the psychological consequences of male infertility,[27] as well as psychological support during waiting times for cancer treatment.[70]

⇒ Appropriate and interactive: the way HCPs communicated the information affected men's feelings; a positive attitude and 'a good mood' among HCPs mirrored less stress in men.[66] Having time to ask questions and interact with HCPs was also appreciated.[32]

⇒ Timely: constant updates of information during their partner's labour and birth were highly appreciated by men. Men who received timely information felt well informed, calm, secured and satisfied.[33 42 52 66] On the other hand, receiving information at inappropriate times was perceived as insufficient.[70]

⇒ Inclusive: involving men in the communication as an equal partner in reproductive healthcare was perceived as necessary.[27]

involvement, to a certain point, since they experienced overinvolvement as a possible sign of controlling behaviour or intimate partner violence.[17] The literature discussed men's involvement and their lack of control and compromised autonomy in various situations in reproductive healthcare, especially during pregnancy and birth. For example, the inability to help or act during their partner's birth made men experience lack of power and control.[46 66 71] Similarly, the uncontrollable process of non-progressing delivery left men with a feeling of helplessness and insecurity.[46] Men appreciated being involved in the decision regarding their partner's elective or emergency caesarean section, but 40% of the men felt they were not involved enough.[69 72] Also, men reported being more in control and more involved in decision-making during an elective caesarean section or normal spontaneous vaginal birth as compared with emergency caesarean section or assisted vaginal birth.[66 69 72] However, they also described situations where they were forced to participate in tasks and rituals without their consent, (e.g. cutting the umbilical cord or touching the child's head before the baby was born).[45]

Even though involvement in decision-making during birth was associated with higher satisfaction,[41 69] it was still important to be able to choose whether to participate or not in different stages of birth.[42]

Compromised autonomy was also reported in the infertility clinic[73] and when banking sperm before cancer treatment.[32] To the contrary, control and involvement in decisions were more satisfactory during home abortions. The pregnant woman made the decision, but the partner's opinion was important for her.[59 64]

### Good treatment increases security and satisfaction

Men wanted HCPs to treat them as persons, respecting their needs, feelings and experiences. HCPs should try to understand the unique situation of each man and take it seriously.[36 42 47 59] Respectful treatment was highly expected and associated with higher satisfaction with care.[53 74] It was especially important to deliver negative news with sensitivity.[43 60] Men who experienced HCPs as professional, empathetic and attentive felt satisfied, important and 'not just a number'.[5 36 59 75] In other cases, men perceived insensitivity and lack of respect or attention in the comments of HCPs, resulting in feeling disappointed and dissatisfied.[41 59 60]

The support of midwives during antenatal, intrapartum and postnatal care was necessary and created a feeling of security and satisfaction. Providing attention and information and addressing men's needs and questions helped men to build trust in the midwives and be supportive to their partners.[42 47 52 66–69] However, men were not always satisfied with the support of midwives, which made men feel insecure, helpless and worried.[41 69 71]

### Confidentiality, a prerequisite to access to SRHC

Confidentiality was considered an essential condition to access certain SRH services, including youth clinics and HIV testing. For example, fear of being recognised in the clinic was one of the main reasons for not being tested for HIV.[76] Getting an HIV test was considered as implicitly disclosing same-sex sexuality, which led to preferring self-testing as an anonymous alternative, especially among non-gay MSM and those who had never been tested for HIV.[77] Therefore, anonymous HIV testing outside the healthcare system was requested and considered helpful for MSM.[78] Similarly, young people visiting youth clinics expressed the importance of HCPs' confidentiality and that they are used to and only work with young people.[5] Trust in HCPs' confidentiality was also described as important in the process of men disclosing victimisation.[29]

### Healthcare system factors

The healthcare system influenced the HCP–user relationship through its effects on HCPs and the healthcare encounters. Among other issues, the literature discussed the organisation of healthcare, the holistic approach (or the lack of it), SRHC as traditionally women-centred care and men's SRHC as 'nobody's mandate'.

### Men's SRH is not a priority

The literature indicated that the clinical training and organisation of care do not give men's SRH enough priority. Nurses, for example, highlighted the lack of basic medical training and organisational support to deal with men's sexual health issues. Their main source of knowledge about men's SRH was received from pharmaceutical companies.[16] Similarly, midwifery education and clinical training do not regularly include andrology, which together with lack of time and organisational support hindered them from providing counselling to men.[17]

Another example of the low priority of men's SRH was the lack of follow-up and continuity of care, which was reported in various services. For example, men reported not being followed up after being prescribed medication for sexual function.[28] Additionally, the stay of men with the family after delivery was not welcomed in some hospitals, even though this was important for men in order to feel supported and to support their new family.[47]

The lack of prioritising men's SRH was also reflected by the few prevention activities that healthcare performs regarding men's SRH. For example, the vast majority of MSM did not encounter any HIV/STI prevention services, despite the importance of making it more available.[63] Another example was the missed opportunity to counsel for sexual health in around one-third of men testing for HIV.[75]

### Lack of holistic care

The literature discussed the lack of holistic care in SRH services. For example, psychological aspects of infertility were usually not acknowledged and therefore overlooked.[36] For couples with repeated pregnancy loss, psychological counselling was restricted to a few with certain criterion and also without considering individual situations.[43] Furthermore, antenatal care was perceived to focus mainly on medical support and rarely on emotional and psychological support, leaving only few users being very satisfied with this aspect of antenatal care.[14 67] Consequently, men who were subjected to gender-based violence were less likely to seek help unless they had severe physical injuries.[29]

### Women-centred reproductive healthcare, a compromised right for inclusion of men

Both men and women expressed a wish to include men and to focus on 'the couple' rather than one partner, that is, equal partners sharing a common reason for visiting reproductive healthcare.[15 36 43 59] Even though men felt that the focus on women in reproductive healthcare is reasonable, they stated that this attention should not exclude men.[36 49] The feeling of exclusion was experienced by men in different reproductive health services, including fertility and antenatal care.[14 15 38 49] One study showed that the investigations and treatments focused only on the women, even when the cause of infertility was a low sperm count, which led to perceive infertility care as the 'women's world'.[15] Additionally, the midwives

discussed sexual and reproductive rights for men as being women's partners rather than being men's own rights, and men's concerns about contraception are dependent on his partner's choice to include him or not in contraceptive counselling.[17]

### Men's SRH is nobody's responsibility

Different HCPs expressed their concern about men's sexual health as 'no one's responsibility'. The attitudes of midwives toward providing counselling to men were divided. Some were positive and found it a continuation of their current responsibilities that concerned women.[17] This opinion was shared by men of pregnant partners who expressed their trust and faith in midwives and saw them as the best ones to promote sexual health among men.[49] Other midwives were reluctant and expressed their difficulty in providing counselling to men. For them, the pregnancy is about the woman's body, and thus, man's participation was not evident.[17] Nurses also questioned if men's sexual health is their duty, especially if it included an emotional aspect. In their opinion, primary care was not equipped to deal with sexual health problems; therefore, they often referred patients to other healthcare units.[16]

### Sociopolitical factors

The outer layer of the model (figure 4) discusses the social and political factors, including social and gender norms, which affect the healthcare system and the attitudes and behaviours of HCPs and users.

### Social and gender norms

The literature described traditional social and gender norms as contributing to setting values for men that hinder their abilities to cope or seek help and affect their sexual health and well-being. For example, infertility was described as a 'malfunction of manhood' that is faced by denial and changes how men perceive their masculinities. The absence of sperm was an identity question and a threat to men's masculinities.[15 27 32 60] Similarly, suffering the decline in sexual function associated with a diagnosis of prostate cancer was perceived as threatening to the male identity and therefore accompanied by feelings of inadequacy and not being a 'real man'.[28 70]

Furthermore, the attempts of men to conform to traditional masculinity norms affected their ability to talk about experiences of violence, especially if they were exposed to intimate partner violence.[29] Additionally, transmen had experiences of vulnerability during gynaecological examination or when they resumed menstrual bleeding after family planning treatment. This was perceived as stressful, humiliating and uncomfortable, as well as a reminder of a sex they 'wanted to forget'.[79]

These 'threatened masculinities' were also reflected through men's health-seeking behaviours. Men disregarded their sexual health, delayed admission of the problem and opted to distance themselves from seeking healthcare.[13] For example, young men had more

difficulties to admit their SRH needs and to seek help as compared with young women.[5] Similarly, the midwives indicated that men only seek help when they have severe symptoms, while also noting that young men are increasingly attending STI testing and are more open to discuss sexual health.[17]

Men expressed increased social expectations on them to be more involved in healthcare during pregnancy and birth, which corresponded to personal willingness and desire to share responsibility for the security and support of their partners.[28 39 45 47] Men were also eager to participate in other reproductive healthcare services, such as infertility treatment and home abortion.[36 64 74] However, men were faced with barriers in their desire to participate and experienced 'paddling upstream' to fulfil their involvement.[40]

The literature also discussed how social and gender norms affect the healthcare system being perceived as women-centred. Youth clinics, for example, were perceived as a place for the SRH of girls, which created a barrier for young men seeking healthcare.[5] Additionally, the social norms hindered HCPs talking about sexual health, especially when the patient is older than the HCP.[16] In turn, HCPs reinforced these social norms by supporting the traditional gender expectations of the woman as the primary infant caregiver and overlooking the importance of shared parenthood and including the man in infant care.[39 80]

Studies also described how social and gender norms affected the way healthcare deals with victimised men. The training and education the emergency departments offered in Sweden about caring for violence victims focus only on women and children and not victimised men.[81] Similar experiences of the reinforcement of traditional gender positions by HCPs were perceived by men subjected to intimate partner violence. These men felt alone since society did not acknowledge their experiences, and the HCPs expected them to embody traditional ideals of masculinities.[29]

## Policies

Politics and policies were rarely discussed in the literature, but there were some mentions of the regulations and guidelines in SRHC, which have been discussed under point 4. The only mention of policy was in the context of gender-based violence. While most of the counties and emergency departments in Sweden had a policy about the care for victims of violence, these policies focus merely on women and children but not men or other groups.[81]

## DISCUSSION

In the previous section, we reviewed, charted and synthesised the available literature in relation to the factors influencing men's experiences in SRHC. To summarise, the majority of the 68 reviewed papers discussed men's experiences in reproductive healthcare, mainly care related to infertility, pregnancy and birth. The literature

lacked men's perspectives on contraception, including condom use and vasectomy. Regarding sexual healthcare, the available literature captured mainly STIs and HIV treatment and prevention but not men's experiences in other sexual health issues, such as impotence or gender-based violence. This focus on STIs and reproduction reflects the biomedical gaze of healthcare, keeping topics like gender-based violence and sexual satisfaction, to a great extent, outside the focus of healthcare and health service research. The literature also lacked the perspectives of particular groups of men who might face different experiences in SRHC, such as transmen, indigenous, national minorities and men with functional variations. Furthermore, MSM were only mentioned in relation to HIV treatment and prevention. Similarly, migrants were the main focus in only two studies related to foreign-born MSM and HIV testing.

The literature indicated that men face difficulties to be included in reproductive healthcare, where they are mostly treated as an accompanying partner, receiving little attention. The knowledge and attitudes of HCPs were crucial for their ability to discuss men's SRH and also for men's experiences in SRHC. Furthermore, the literature rarely discussed healthcare organisation and policies and how they affect men's health-seeking behaviours and experiences in SRHC. Lastly, men's right to SRH is usually not stressed in the literature, unless it is related to a specific group of men, such as MSM and transmen.

While we presented the factors influencing men's experiences in SRHC in separate levels and the reviewed articles did not explicitly study the interaction between these levels, the theoretical framework still enables us to understand the interaction between these determinants. We presented some examples in the results of how these levels are linked and influence each other. The interaction between gender and social norms with the other determinants might be of special significance. For example, the literature described how traditional social and gender norms affect the attitudes and behaviours of HCPs. A clear example of this was how men were treated as an accompanying partner during healthcare visits related to infertility, antenatal care and birth. In many cases, men are still not seen as an equal partner or as a primary caregiver for their newborns, which likely influences the attitudes of HCPs toward men seeking antenatal care, in turn affecting men's experiences of those services negatively.[14 40]

While traditional gender norms and values of masculinities provide important pieces in explaining men's health-seeking behaviours, a more comprehensive picture of men's experiences in SRHC is needed. The literature showed other determinants of men's experiences in SRHC, including how the healthcare system is organised. It seems SRHC in the Nordic countries focuses mainly on women, while there is a lack of knowledge about men's SRH and no clear entry for men into SRHC. The healthcare system should adapt a gender-responsive approach that ensures accessible healthcare services for men and

which through its approach addresses the impacts of gender norms on men, women and HCPs.[2 82]

To reach universal access to SRHC and gender equity, it is of importance to engage men in SRH and ensure that their needs are met.[83] Improving men's experiences in SRHC in the Nordic countries is not only important for improving men's SRH but also could enable men to strengthen their support of women's SRH and thus gender equality.[83 84] Meeting men's needs for SRHC could consequently decrease STIs and unintended pregnancies, and improve parenting and family relationships.[9 21] Such specific focus on men in the SRHC organisation to improve men's health and rights with the goal to contribute to gender equality will benefit both men and women.[85 86]

## Strengths and limitations of this study

To the best of our knowledge, this review is the first to examine the experiences of men in SRHC in the Nordic countries. The review provides interesting and important information about these experiences, by organising them in a theoretical framework that makes it easier to understand and draw conclusions. However, the design of our study and our search terms are best suited to draw conclusions about men's experiences of SRHC rather than the determinants of SRHC utilisation even though we reported both in this study.

Even though we developed and followed our search strategies thoroughly, the review has some limitations. The broad nature of the field and the wide variety of terms related to SRH make it difficult to assure the inclusion of all relevant literature. Additionally, due to the restricted time of the project and the limited funding, we included only peer-reviewed literature in two databases, we did not register a review protocol prior to the study and no stakeholder consultation was conducted after performing this scoping review. However, we complemented the search with manually screening the reference lists of the identified literature. Another strength of this review was the use of a Nordic-specific database without restriction to language, which ensured an equal inclusion of the literature from other Nordic countries, even though most of the literature in this review was published in Sweden.

Furthermore, the adapted framework allowed us to use a relevant ecological lens on men's experiences in SRHC and to systematically identify and categorise the concepts discussed in the selected literature. However, the use of this framework might have caused us to overlook aspects of the research topic that fell outside the interest of this scoping review.

Finally, it is important to note that, when reporting results and discussing them, we choose to implicitly treat the Nordic countries as essentially similar. While we argue that this makes sense because of the actual similarities between these countries and their healthcare systems, we also acknowledge that this might obscure important differences between or within countries (eg, relation to place of residence (rural vs urban) or cultural differences).

## CONCLUSION

Despite the uncontroversial importance of men's right to access SRHC on equal terms, the available literature indicated that SRH is mainly the domain of women and healthcare around men's SRH is not sufficiently prioritised. A more comprehensive picture of men's experiences in SRHC is needed.

There is a lack of knowledge about men's SRH and no clear entry for men into SRHC. This indicates the necessity for improvements in the medical education of HCPs and in health system interventions. Further research should examine the influence of policies and the healthcare organisation on men's access and experiences in SRHC and explore the identified knowledge gaps of men's experiences in SRHC related to specific groups of men such as migrants, MSM and transmen and to specific SRH subject areas such as sexual function, contraceptive use and gender-based violence.

**Contributors** MB was the lead reviewer and author of the manuscript. A-KH and JPS contributed to the identification and selection of articles and contributed to data interpretation. HB and KE contributed to data interpretation and provided comments on the manuscript. All authors agreed on the final version of the manuscript. A-KH was the project leader.

**Funding** This study was supported by the Public Health Agency of Sweden (Folkhälsomyndigheten).

**Disclaimer** The Public Health Agency of Sweden was not involved in the analysis, interpretation, writing or the decision to submit this paper for publication.

**Competing interests** None declared.

**Patient consent for publication** Not required.

**Provenance and peer review** Not commissioned; externally peer reviewed.

**Data availability statement** Data sharing not applicable as no datasets generated and/or analysed for this study. No primary data were collected for this review.

**ORCID iD**
Mazen Baroudi http://orcid.org/0000-0002-0609-8745

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
