## [Reviewer comments · BMJ Open]

ARTICLE DETAILS

TITLE (PROVISIONAL)	Men and sexual and reproductive healthcare in the Nordic countries: a scoping review
AUTHORS	Baroudi, Mazen; Stoor, Jon Petter; Blåhed, Hanna; Edin, Kerstin; Hurtig, Anna-Karin

VERSION 1 – REVIEW

REVIEWER	Maria Lohan Queens University of Belfast
REVIEW RETURNED	07-Jun-2021

GENERAL COMMENTS	This is an original rigorous and significant scoping review which will be a very valuable addition to the literature. My main recommendation to the authors would be to consider a tension the in the aim of the paper as follows and subsequently some of the framing of the results: The paper makes the case that 'men's experiences of SRHC' are not adequately reviewed in the literature. The paper finds up to 68 studies worthy of inclusion. 61/68 address men's experiences and so this review is well set up to synthesize these studies and make a very positive contribution. However, the authors set the aim as 'to identify knowledge gaps and factors influencing men regarding SRHC in the Nordic countries' Hence the results move away from examining men's experiences to also examining the predictors of men's use. While the results on predictors of/socio-demographic factors associated with men's use of HIV testing is interesting - it is not clear that the body of the research allows reliable conclusions on this. The body of work is most clearly set up to report men's experiences. Similarly when talking about predictors of prostate cancer care, the authors use the word 'studies' but report only one study: 'Studies showed no associations between age and the overall satisfaction with cancer care, while a higher level of education was associated with lower overall satisfaction with prostate cancer care.⁶⁵ Furthermore, the literature indicated that manual workers were less likely to receive a bone scan and radical prostatectomy, and they had higher overall and cancer-specific mortalities as compared to non-manual employees.⁶⁶ The four-part framework from Kilbounre et al is useful to report the experiences and the factors influencing experiences.
--

	So in summary overall, I would like to suggest to authors that they consider limitations of their conclusions relating to sociodemographic predictors - and the strength of their review in synthesizing literature on men's experiences in Nordic countries in relation to SRHC. A second more minor recommendation is to reconsider the examples of sex differences given in the introduction. Women also underreport sexual dysfunction. It is also not universally the case that women test more often for STIs than men. The authors do not make the claim that it is universally the case but the examples imply when that context is not given. Could the case be made for the importance of looking at men's sexual and reproductive health alongside that of women - and especially perhaps that men's reproductive needs are neglected?
--	---

REVIEWER	Nakia Lee-Foon University of Toronto
REVIEW RETURNED	13-Jul-2021

GENERAL COMMENTS	Abstract Page 2, Line 31-it is unclear as to what 'no assigned profession' is referring to. Would recommend revising and providing an explicit example of a profession that is being referred to here. Page 2, line 37- It is unclear what exactly the 'identified knowledge gap' is. Would recommend stating this for greater clarity. Strengths and limitations of the study, pg. 2-3 It would also be interesting to also note that this study does not find many in depth, articles on differences in men's experiences based on their gender identity (cisgender and transgender men), immigration status as other, international studies indicate variations in experiences based on these and other social locations. Introduction Page 4, Lines 15-16. This is an interesting point. However, there is a lot of international literature focused on MSM's SRHC use/experiences accessing SRHC and their use/experiences often vary greatly from cisgender men's. For greater clarity, it would be beneficial to provide a brief explanation/example of MSM's SRHC's more frequent use. Page 4, line 27. 'This mirrors the lack of response of the health system to men's needs . . .' This is a bit unclear as to what 'this mirrors' is in reference to. Would recommend rephrasing for greater clarity -While MSM are mentioned early into the introduction section, they are only mentioned once, making the article appear to predominantly focus on cisgender, heterosexual men. If, this is in fact the focus, this should be explicitly stated in the introduction. Without this statement, the reader will expect to see discussion about MSM and their SRHC experiences in Nordic countries.
--

	Methods 1) The paper's title points to the use of a scoping review to locate selected articles. However, not all parts of the scoping review's methodological framework were noted (e.g. research question), no explanation as to why only 2 databases used and why only peer-reviewed empirical studies were assessed and a grey literature search avoided. This section would benefit from briefly responding to the abovementioned items. Further, scoping reviews have the option of a consultation exercise where one can go to stakeholders to seek additional feedback about the review findings. If space permits, it would be good to state if this did or did not happen. 2) It is also unclear if the eligibility criteria included all articles irrespective of the language it was written in. For greater clarity, it should be written here and not just in the results section. Data extraction and synthesis Page 5, line 28. There appears to be a grammatical error '... and the result parts ...' would recommend revising. -It would also be beneficial to explain why the WHO's framework was used here vs another. Results This section provides a good overview of the selected studies. As Nordic countries have had increases in their immigrant populations in recent years, with these populations often experiencing healthcare delivery and access in ways that often differ from those born in these countries, it would be beneficial to briefly discuss if any of the selected papers specifically focus on these populations. Theoretical framework for Analysis -it's unclear why Kilbourne's framework was used and adapted here. Would benefit from briefly discussing its use and why. Page 7, line 16. 'The literature described on how factors ...'. Would recommend rephrasing section for greater clarity Page 7, lines 30-34. Would recommend not using 'Additionally' to start to subsequent sentence. Would recommend modifying this for greater readability. 2.1 Prevention and Control of HIV and other STIs Page 8, line 30-31. It's unclear as to what 'it' refers to at the end of this sentence. Would recommend explicitly stating what 'it' is for greater clarity. -It's unclear what age group is being referred to here when the term 'school boys' is being used. Would recommend being explicit by what age group is being referred to here. -was there any variations in the literature for MSM vs heterosexual, cisgender men? If yes it would greatly enrich this section if this were briefly stated. BOX 1. The sociodemographic factors of users in relation to HIV testing and treatment
--	--

	-it's unclear how knowledge and risk perception are considered sociodemographic factors in this context as they're not sociological or demographic characteristics of a group/(s). If this box remains, would recommend rephrasing this as key characteristics of users in relation to HIV testing and treatment to avoid any confusions. BOX 2. The characteristics of satisfying information and communication-men's views Page 11. Lines 11-12. It's unclear what 'exaggerations', 'exaggerated information' means, would recommend rephrasing/explaining for greater clarity. Discussion -This section provides a robust critique of the current state of SRHC literature focused on men. While it is appreciated that the authors critique the lack of particular groups discussed in the literature (e.g., national minorities, MSM, transgender men) it would greatly enhance this section if a sentence or so was added to explain why this literature gap is significant as well as the seeming focus of the literature on men's reproductive—fertility vs infertility compared to other aspects of sexual health (e.g., satisfying sex life, the choice to reproduce). Strengths and limitations of the study Page 16, lines 22-23. See point 1 in methods section feedback. Page 16, line 29. It's stated that a relevant ecological lens was used but it was not clearly stated/implied in earlier sections of this article. If it was, it needs to be more explicit for the reader/named for greater clarity. Conclusion -Conclusion provides a good summary of the review and potential future next steps. Would also recommend further emphasizing the need for additional research that moves beyond a cis-gender, heteronormative view of SRHC in Nordic countries.
--	---

VERSION 1 – AUTHOR RESPONSE

Reviewer: 1

Ms. Maria Lohan, Queens University of Belfast

This is an original rigorous and significant scoping review which will be a very valuable addition to the literature. My main recommendation to the authors would be to consider a tension the in the aim of the paper as follows and subsequently some of the framing of the results: The paper makes the case that 'men's experiences of SRHC' are not adequately	Thank you for accepting to review our paper. Your time and efforts are much appreciated. We agree with you that our scoping review is best suited to answer the question about men's experiences in SRHC more than identifying the factors influencing men's use of SRHC which is also reflected by our search terms.
--	---

reviewed in the literature. The paper finds up to 68 studies worthy of inclusion. 61/68 address men's experiences and so this review is well set up to synthesize these studies and make a very positive contribution. However, the authors set the aim as 'to identify knowledge gaps and factors influencing men regarding SRHC in the Nordic countries' Hence the results move away from examining men's experiences to also examining the predictors of men's use. While the results on predictors of/socio-demographic factors associated with men's use of HIV testing is interesting - it is not clear that the body of the research allows reliable conclusions on this. The body of work is most clearly set up to report men's experiences. Similarly when talking about predictors of prostate cancer care, the authors use the word 'studies' but report only one study: 'Studies showed no associations between age and the overall satisfaction with cancer care, while a higher level of education was associated with lower overall satisfaction with prostate cancer care.⁶⁵ Furthermore, the literature indicated that manual workers were less likely to receive a bone scan and radical prostatectomy, and they had higher overall and cancer-specific mortalities as compared to non-manual employees.⁶⁶' The four-part framework from Kilbounre et al is useful to report the experiences and the factors influencing experiences. So in summary overall, I would like to suggest to authors that they consider limitations of their conclusions relating to sociodemographic predictors - and the strength of their review in synthesizing literature on men's experiences in Nordic countries in relation to SRHC.	Access to healthcare is however a multi-stage process that includes perceiving the need to seek healthcare, seeking and visiting healthcare, and the outcome of healthcare. We think that each step of this process influence men experiences in healthcare. Therefore, we chose to also synthesize the literature we found in relation to the utilization of healthcare and the outcome of healthcare even though –as you noted- no conclusion is made from this body of literature, As a response to your valuable comment and to make the readers more aware about this issue, we have now addressed this in the strengths and limitation of the study: “However, the design of our study and our search terms are best suited to draw conclusions about men’s experiences of SRHC rather than the determinants of SRHC utilization even though we reported both in this study.”(Lines 487 to 489)
A second more minor recommendation is to reconsider the examples of sex differences given in the introduction. Women also underreport sexual dysfunction. It is also not universally the case that women test more often for STIs than men. The authors do not make the claim that it is	Thanks for pointing this out. Indeed, the examples given might implicitly indicate that women have no SRH problems, therefore adding the following clarification was necessary. “Addressing men’s sexual and reproductive health (SRH) needs alongside that of women’s is

universally the case but the examples imply when that context is not given. Could the case be made for the importance of looking at men's sexual and reproductive health alongside that of women - and especially perhaps that men's reproductive needs are neglected?	essential, however men's SRH is neglected. "(Lines 58, 59)
--	---

Reviewer: 2

Dr. Nakia Lee-Foon, University of Toronto

This paper provides an overview of an important aspect of men's SRHC that is often overlooked in the literature. It highlights significant service delivery gaps and points to areas where men's SRHC can be improved. Kindly see attached for my feedback, which I hope, will further enhance this important scoping review.	We would like to appreciate your time and efforts spent on this review. We have tried to respond to your important comments as follows:
Abstract	
Page 2, Line 31-it is unclear as to what 'no assigned profession' is referring to. Would recommend revising and providing an explicit example of a profession that is being referred to here.	Thanks for pointing this out. A clarification has been added as follows: "no assigned healthcare profession for men's SRH issues" (Lines 35, 36)
Page 2, line 37- It is unclear what exactly the 'identified knowledge gap' is. Would recommend stating this for greater clarity.	Thanks for this comment. We have now added which knowledge gap is identified as follows: "The literature lacked the perspectives of specific groups of men such as migrants, MSM and transmen and the experiences of men in SRHC related to sexual function, contraceptive use and gender-based violence besides a gap regarding the influence of policies and healthcare organization on how men perceive SRHC. These knowledge gaps, taken together with the lack of a clear entry point for men into SRHC indicate the necessity of an improved health and medical education of healthcare providers, as well as of health system interventions." (Lines 39 to 43)
Strengths and limitations of the study, pg. 2-3	
It would also be interesting to also note that this study does not find many in depth, articles on differences in men's experiences based on their gender identity (cisgender and transgender men), immigration status as other, international studies indicate variations in experiences based on these	We agree with you that the study does not find many in-depth studies about the experiences of various men's groups. We consider this, however, as a limitation of the literature rather than a limitation of our study. We tried to be inclusive to all men's experiences and

and other social locations.	did not restrict our search term to cis-men. Therefore we chose not to mention this limitation here but in the discussion. “The literature also lacked the perspectives of particular groups of men who might face different experiences in SRHC, such as transmen, Indigenous, national minorities and men with functional variations. Furthermore, MSM were only mentioned in relation to HIV treatment and prevention. Similarly, migrants were the main focus in only two studies related to foreign-born MSM and HIV testing.” (Lines 445 to 449)
Introduction	
Page 4, Lines 15-16. This is an interesting point. However, there is a lot of international literature focused on MSM’s SRHC use/experiences accessing SRHC and their use/experiences often vary greatly from cisgender men’s. For greater clarity, it would be beneficial to provide a brief explanation/example of MSM’s SRHC’s more frequent use.	Thanks for your comment. We agree with you that this statement is too general and would benefit from some nuances and clarifications. We have now rephrased this statement as follows: “However, various groups of men might have different health seeking behaviours and different experiences in SRHC. For example, men with high socioeconomic status and men who have sex with men (MSM) seek SRHC more often. The higher use of SRHC among MSM might be due to their higher needs. Furthermore, MSM experiences in SRHC might differ due to their level of openness about their sexual orientation and related to structural factors such as homophobia.”(Lines 66 to 70)
Page 4, line 27. ‘This mirrors the lack of response of the health system to men’s needs . . .’ This is a bit unclear as to what ‘this mirrors’ is in reference to. Would recommend rephrasing for greater clarity	Thanks for pointing that out. This have been rephrased as follows: “These experiences might be due to the lack of response of the health system to men’s needs that can be related to healthcare organization and delivery”(Lines 78 to 80)
-While MSM are mentioned early into the introduction section, they are only mentioned once, making the article appear to predominantly focus on cisgender, heterosexual men. If, this is in fact the focus, this should be explicitly stated in the introduction. Without this statement, the reader will expect to see discussion about MSM and their SRHC experiences in Nordic countries.	We intended to be inclusive to all men’s experiences and did not exclude transmen or non-heterosexual men from our search terms. To the opposite, our search term included, as attached in appendix 1, the following terms: “Homosexuality, Male[mesh] OR “MSM” OR “men having sex with men”[TIAB] OR “men who have sex with men”[TIAB] OR “men who have

	sex with other men”[TIAB] OR ((transgender*[TIAB] OR transgender persons[MH] OR transsexual*[TIAB]) AND man) OR transman[TIAB] OR “trans men”[TIAB] OR transmen[TIAB])” We found only 2 articles specific for transmen experiences and 9 articles dealing with MSM (all about sexually transmitted diseases. These experiences were mentioned in the result section when suitable trying to keep the balance with other men’s groups. We agree that the lack of literature about MSM and transmen is remarkable, therefore, we highlighted this in the first paragraph of the discussion and in other places as per response to other valuable comments (see Lines 39 to 43, Lines 66 to 70, Lines 445 to 449 and Lines 515 to 519)
Methods	
1) The paper’s title points to the use of a scoping review to locate selected articles. However, not all parts of the scoping review’s methodological framework were noted (e.g. research question), no explanation as to why only 2 databases used and why only peer-reviewed empirical studies were assessed and a grey literature search avoided. This section would benefit from briefly responding to the abovementioned items. Further, scoping reviews have the option of a consultation exercise where one can go to stakeholders to seek additional feedback about the review findings. If space permits, it would be good to state if this did or did not happen.	Thanks for your valuable comments. To increase the clarity of the method section, we have added some clarifications as follows: “This review was performed according to Arksey and O’Malley’s method stages for conducting a scoping review, which includes identifying the research question, literature search, study selection, charting and synthesizing. The research questions included: 1) What is the current status of the literature published in Scandinavian regarding men and SRHC? 2) How are men in the Scandinavian countries experiencing SRHC?”(Lines 93 to 97) We also included a statement in the strengths and limitations section as follows: “Due to the restricted time of the project and the limited funding we included only peer-reviewed literature in two databases, we did not register a review protocol prior to the study and no stakeholder consultation was conducted after performing this scoping review.”(Lines 492 to 495)
2) It is also unclear if the eligibility criteria included all articles irrespective of the language it was written in. For greater clarity, it should be	This have now been added to the method section:

written here and not just in the results section.	"A structured search of the literature was conducted using two databases, PubMed and SveMed+ (a Scandinavian database) without restriction of language."(Lines 99, 100)
Data extraction and synthesis	
Page 5, line 28. There appears to be a grammatical error ' . . . and the result parts . . . ' would recommend revising.	Thanks for pointing this error out. This have now been corrected.
-It would also be beneficial to explain why the WHO's framework was used here vs another.	A clarification has been added to the text as follows: "This framework was used because it demonstrates the interlinked nature between sexual health and reproductive health, yet clearly distinguish topics for intervention and research in both sexual health and reproductive health"(Line 140 to 142)
Results	
This section provides a good overview of the selected studies. As Nordic countries have had increases in their immigrant populations in recent years, with these populations often experiencing healthcare delivery and access in ways that often differ from those born in these countries, it would be beneficial to briefly discuss if any of the selected papers specifically focus on these populations.	We found only two papers that focus on foreign-born MSM and their experiences with HIV testing. The lack of focus on this group is mentioned in the discussion as follows: "The literature also lacked the perspectives of particular groups of men who might face different experiences in SRHC, such as transmen, Indigenous, national minorities and men with functional variations. Furthermore, MSM were only mentioned in relation to HIV treatment and prevention. Similarly, migrants were the main focus in only two studies related to foreign-born MSM and HIV testing."(Line 445 to 449)
Theoretical framework for Analysis	
-it's unclear why Kilbourne's framework was used and adapted here. Would benefit from briefly discussing its use and why.	We have now motivated the use of Kilbourne et al. framework as follows: "Kilbourne et al. framework provides a multi-level approach to understand healthcare disparities. It provides an ecological lens that goes beyond individual to interpersonal and organizational factors."(Line 164 to 166)
Page 7, line 16. 'The literature described on how factors . . . '. Would recommend rephrasing section for greater clarity	Thanks for pointing out this error. This has been corrected.
Page 7, lines 30-34. Would recommend not using 'Additionally' to start to subsequent sentence.	Thanks for your comment. The connecting words

Would recommend modifying this for greater readability.	have now been changed.
2.1 Prevention and Control of HIV and other STIs	
Page 8, line 30-31. It's unclear as to what 'it' refers to at the end of this sentence. Would recommend explicitly stating what 'it' is for greater clarity.	Thanks for pointing this out. This now reads: "Most of the literature focused on HIV testing, treatment and their sociodemographic determinants"(Line 227)
-It's unclear what age group is being referred to here when the term 'school boys' is being used. Would recommend being explicit by what age group is being referred to here.	This has now been specified "the attitudes of upper secondary school boys (median age=18) toward HPV vaccination"(Lines 229, 230)
-was there any variations in the literature for MSM vs heterosexual, cisgender men? If yes it would greatly enrich this section if this were briefly stated.	Thanks for your valuable comment. Most of the studies found on STI and HIV are MSM. The few studies where the main focus is not MSM are related to determinant of the utilization of services rather than experiences in healthcare. Therefore, we think that no conclusions could be drawn about the variation between MSM vs. hetero or trans vs. cis.
BOX 1. The sociodemographic factors of users in relation to HIV testing and treatment	
-it's unclear how knowledge and risk perception are considered sociodemographic factors in this context as they're not sociological or demographic characteristics of a group/(s). If this box remains, would recommend rephrasing this as key characteristics of users in relation to HIV testing and treatment to avoid any confusions.	Thanks for your suggestion. We have changed the box title to "Key characteristics of users in relation to HIV testing and treatment"(Box 1)
BOX 2. The characteristics of satisfying information and communication-men's views	
Page 11. Lines 11-12. It's unclear what 'exaggerations', 'exaggerated information' means, would recommend rephrasing/explaining for greater clarity.	We have rephrased and added clarification as follows: "Contradictions, unrealistic information and lack of reliable information caused frustration. Exaggerated information (i.e. under- or overstating the real situation) was associated with unease, confusion and a sense of not being taken seriously"(Box 2)
Discussion	
-This section provides a robust critique of the current state of SRHC literature focused on men.	Thanks for pointing this out. We have now added some clarification discussing the gap in the

While it is appreciated that the authors critique the lack of particular groups discussed in the literature (e.g., national minorities, MSM, transgender men) it would greatly enhance this section if a sentence or so was added to explain why this literature gap is significant as well as the seeming focus of the literature on men’s reproductive—fertility vs infertility compared to other aspects of sexual health (e.g., satisfying sex life, the choice to reproduce).	literature as follows: “ This focus on STIs and reproduction reflects the biomedical gaze of healthcare. Keeping topics like gender-based violence and sexual satisfaction, to a great extent, outside the focus of healthcare and health service research. The literature also lacked the perspectives of particular groups of men who might face different experiences in SRHC, such as transmen, indigenous, national minorities and men with functional variations. Furthermore, MSM were only mentioned in relation to HIV treatment and prevention. Similarly, migrants were the main focus in only two studies related to foreign-born MSM and HIV testing.” (Lines 443 to 449)
Strengths and limitations of the study	
Page 16, lines 22-23. See point 1 in methods section feedback.	This part reads now as follow: “Additionally, due to the restricted time of the project and the limited funding we included only peer-reviewed literature in two databases, we did not register a review protocol prior to the study and no stakeholder consultation was conducted after performing this scoping review”(Lines 492 to 495)
Page 16, line 29. It’s stated that a relevant ecological lens was used but it was not clearly stated/implied in earlier sections of this article. If it was, it needs to be more explicit for the reader/named for greater clarity.	Thanks for this comment. We have now introduced the “ecological lens” concept in the result section while motivating the use of Kilbourne et al. framework: “Kilbourne et al. framework provides a multi-level approach to understand healthcare disparities. It provides an ecological lens that goes beyond individual to interpersonal and organizational factors.”(Lines 164 to 166)
Conclusion	
-Conclusion provides a good summary of the review and potential future next steps. Would also recommend further emphasizing the need for additional research that moves beyond a cis gender, heteronormative view of SRHC in Nordic countries.	Thanks for this recommendation. We have now added this to the conclusion as follows: “Further research should examine the influence of policies and the healthcare organization on men’s access and experiences in SRHC and explore the identified knowledge gaps of men’s experiences in SRHC related to specific groups of men such as migrants, MSM and transmen and to specific SRH subject areas such as sexual

	function, contraceptive use and gender-based violence.”(Lines 515 to 519)
--	--